



# Demonstration of a Fault Impact Reduction Control Module for Wind Turbines

Benjamin Anderson*[1] and Edward Baring-Gould[1]

[1]National Renewable Energy Laboratory, 15013 Denver W Pkwy, Golden, CO, USA

**Correspondence:** Benjamin Anderson (benjamin.anderson@nrel.gov)

**Abstract.** Traditionally, wind turbines in distribution applications make control decisions as isolated systems. They generally provide maximum power output during operation and manage internal faults with little consideration of the rest of the power system. Although fault detection and tolerance schemes are widely researched and implemented, controls to ameliorate such faults are uncommon in research and industry. The rapid shutdown of a wind turbine in a large transmission-connected wind

plant will have a minimal impact on a large power system, but in a microgrid or isolated grid context the abrupt loss of a single wind turbine may cause grid instability and high stress on the system. This paper demonstrates a fault impact reduction control (FIRC) module for a wind turbine, which implements wider warning thresholds around fault thresholds. When the turbine crosses a warning threshold, the controller sends its predicted action to the grid controller, which facilitates the grid operator's response to a potential wind turbine fault, and then takes appropriate action to ameliorate the fault. Various test cases

demonstrate the controller action under a variety of faults, and various scenarios demonstrate the grid benefit of an FIRC in both microgrid and grid-connected contexts. The FIRC maximizes wind turbine generation and eases generation transition under a variety of fault scenarios. The FIRC module is easy to integrate with the existing controller, and can be easily modified to include the various warnings and thresholds that the user desires. This analysis is mainly performed in a MATLAB-Simulink-based research wind turbine model and is also implemented in the existing LabVIEW-based controller of the same research

turbine at NREL.

## 1   Introduction

Traditionally, wind turbines in distribution applications have made control decisions as isolated systems. In the last few years, distributed wind turbines have been required to ride through and respond to external grid conditions per IEEE standard 1547-2018 (noa, 2018), but no such standard exists for internal faults. They have generally provided maximum power output during

operation, and shut down without warning when experiencing internal faults. Although the unexpected shutdown of a single grid-connected wind turbine is largely ameliorated by the other turbines in a large wind plant and other generators on the grid, shutdown caused by faults are still a source of economic loss. There are a variety of factors contributing to this loss. First, when a wind turbine faults, it no longer produces energy. Second, it has to be replaced by other reserve generation, often at a high cost. Third, a faulted turbine may require an expensive repair to resume operation, and repeated shutdowns may necessitate

further maintenance.





In smaller systems with a less diverse generation portfolio, such as an isolated system or an islanded microgrid, an unexpected loss of any single generator becomes more critical, as a single wind turbine may represent a large portion of generation capacity. As such, wind turbines across all deployment types are urged to become more capable of riding through internal turbine faults. This article demonstrates a novel wind turbine fault impact reduction control (FIRC) module and outlines its

benefits through the desktop simulation of some common wind turbine faults.

## 1.1    Common Wind Turbine Faults

There are a variety of sensors monitoring various operational parameters within a wind turbine such as temperature, vibration, torque, and rotation rates. A fault within the wind turbine will occur when a sensor reading exceeds a certain bound, which traditionally causes the turbine to shut down immediately, either through a normal or emergency stop. Faults can occur in

all wind turbine subsystems and have a variety of causes, including mechanical system failures, electrical system failures, supervisory control and data acquisition (SCADA) failures, and sensor failures. The European Union (EU) Reliawind project documents various types of intrinsic fault categories and corresponding downtimes (Wilkinson and Hassan, 2011). As shown in Fig. 1, the pitch system and electric system account for half of wind turbine downtime as a result of failures. Downtime and failure rate are unequal because different failures tend to take different amounts of time to fix. The example faults simulated in

this paper may have root causes spanning the electric system, sensors, control system, pitch system, generator, and drive train. However, they are selected for ease of display of FIRC module functionality, not for rate of incidence. European wind turbine reliability campaigns have shown a fault rate caused by technical failure of one to two faults per turbine year (Sheng, 2013), spanning a variety of wind turbine subsystems.

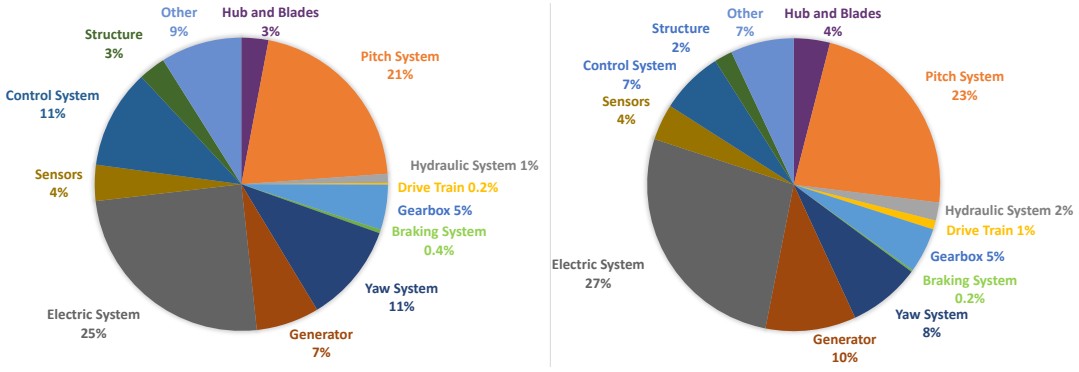

**Figure 1.** Failure rate (left) and downtime (right) as a result of various turbine subsystems. Figure generated from Reliawind report data (Wilkinson and Hassan, 2011).

## 1.2    Fault Controllers in Literature

There are many papers published on fault control, and the most recent literature review of the topic, written by Habibi et al., provides an excellent summary of the body of research (Habibi et al., 2019). Most of the literature details fault detection and





isolation (FDI) schemes, whereas relatively fewer sources present fault tolerant control (FTC) schemes. FDI identifies and isolates sensor deviations: determining which component has the deviation, its severity, and its type. FTC seeks to maintain stable, robust, nominal wind turbine operational setpoints, such as generator torque, power, or rotation speed, in the midst of sensor deviations that come from a variety of sources, as found by FDI. The wind turbine is a complex system with many subsystems containing mechanical, electrical, and sensor components. As such, FDI is critical to FTC, as it identifies the specific sensor deviations that FTC responds to. If sensor deviations detected by FDI cross fault thresholds, the wind turbine shuts down.

There are three main FDI approaches: signal-based, model-based, and physical redundancy (Habibi et al., 2019). Signal-based approaches categorize signal deviations, from, for example, vibration, acoustic, or temperature sensors, and are reviewed by (Hameed et al., 2009; Salem et al., 2014; Tchakoua et al., 2014; Odgaard and Stoustrup, 2013; Qiao and Lu, 2015a, b; García Márquez et al., 2012; Wilkinson et al., 2014). Model-based approaches only utilize control loop input/output signals to detect deviations, (Pourmohammad and Fekih, 2011)) and are reviewed well by Habibi et al. (2019). Because they map specific component response to stimuli, they are good at rejecting noise and system glitches that could be interpreted as faults in other FDI types. The final FDI type is physical redundancy, in which deviant sensors are bypassed by other operational sensors (Habibi et al., 2019). This article will not focus on FDI, as the research turbine considered is already heavily instrumented with a custom FDI system, that uses both signal-based and model-based approaches.

There are two main FTC approaches: active and passive. Passive FTC is optimized for the nominal case, but has leeway that allows for operation under sensor deviations. Active FTC responds to sensor deviations, and generally has better performance (Sloth et al., 2010). Two prominent active methods are virtual sensor/actuator (VSA) and controller reconfiguration. In VSA, a virtual sensor corrects the signal from a broken sensor to remove the deviant effect (Rotondo et al., 2012). It is simpler than controller reconfiguration, increasing its attractiveness to industry. In controller reconfiguration, the baseline controller is reconfigured under deviant conditions to guarantee stability and good performance. It may switch to a new controller, modify controller parameters, or use hardware/software redundancies (Simani and Castaldi, 2012; Sami and Patton, 2012).

Odgaard has written several papers on the testing of FTC methods. He created a benchmark model for simulating deviation detection and response with models of wind turbine subsystems consisting of parameters that could be adjusted to simulate deviations, (Odgaard et al., 2009) and made it more realistic (Odgaard and Johnson, 2013) by integrating the widely used FAST code developed at the National Renewable Energy Laboratory (NREL) (noa, 2021). He tested various leading FDI (Odgaard et al., 2013) and FTC (Pourmohammad and Fekih, 2011) methods, and found VSA to be the best method for power command tracking.

Most of the reviewed literature details and analyzes methods for detecting sensor deviations and maintaining operation, typically at the maximum power point or rated power, despite those deviations. Deviant sensors and components are typically bypassed by FTC, to allow the wind turbine to ride through the sensor deviations until a fault condition arises, necessitating immediate shutdown. The assumption is that the sensor deviations are not severe enough to require shutdown, (e.g., a broken sensor but no mechanical or electrical issue in the wind turbine itself), and that continued stable, robust operation is both possible and the best course of action. However, little literature was found on control action to ameliorate sensor deviations



approaching serious fault conditions, or fault prediction based on sensor data. One recent and focused example of FTC advancements that could be expanded on, by Liu et al. (2021), developed an FTC for floating offshore wind turbines that deals with blade and actuator faults, both for severe (stuck pitch actuator) and non-severe (degraded pitch actuator) faults, based on

a subspace predictive repetitive control approach. Wind turbine pitch control has two general goals: producing steady, maximum power and alleviating blade loads. This FTC does this while taking faults into account, with two main features: first, the FTC mitigates fault-induced loads by creating control laws adapted to the faulty condition. Second, FTC typically requires increased pitch action, so this FTC restricts blade excitation at specific frequencies (1P and 2P) to reduce loads, and to reduce the wind turbine's excitation by external random noise. This reduces fatigue loading and sudden surges in rotor power. However,

this paper was solely focused on pitch actuator and blade faults, and did not communicate predictive fault information to an operator.

When a turbine approaches fault conditions, instead of overriding sensor deviations and continuing at full power which could further exacerbate a fault, it may be prudent to modify the operational set points to mitigate the fault condition. Fault prevention is important, as faults can cause costly downtime and potential grid instability, especially in isolated or microgrid

energy systems with a high wind energy contribution. Also, providing microgrid operators with predictive wind turbine action facilitates load balancing and generation transitions. Finally, an easy-to-integrate control module that can be added to already-deployed wind turbines and can be modified to meet the safety needs of any specific scenario provides benefit to all systems.

The FIRC module developed here addresses these gaps (ease of integration, predictive information, and fault mitigation) with a focus on mitigating fault conditions and providing predictive information to the grid operator while still producing as

much power as possible. Similar to FTC, it uses information gathered from a separate FDI system to make its control decisions. However, the FDI system itself is not within the scope of the FIRC module, which can be easily modified to consider whatever faults the user desires, depending on their use case. The FIRC module can be easily added on to any existing wind turbine control system, and does not replace the existing controller, whether an FTC or otherwise.

### 1.3 Fault Impact Reduction Control Module

This section outlines the development of the FIRC, wherein the wind turbine enters a warning state when a sensor approaches a fault threshold, and communicates the current turbine operational mode, projected time until an event occurs, and power level that will be targeted when said event occurs to the grid operator. The turbine controller also implements changes to ameliorate a warning, instead of waiting for the wind turbine to shut down completely. Warnings do not override faults. When a fault threshold is crossed, the wind turbine still shuts down in the determined manner. The proposed FIRC provides the following

benefits to the existing controller it supplements:

1. Gives the grid operator more time to react to potential wind turbine faults with its dispatch decisions

2. Reduces the amount of spinning reserve and number of other generator starts needed to offset wind turbine faults

3. Maximizes the amount of power that the wind turbine can provide under warning conditions





4. Reduces the risk of safety hazards caused by operating near fault conditions.

5. Is easily implemented and modified to adapt to changing wind turbine system needs.

These benefits are significant in grid-connected settings, but are most important in microgrids and isolated grids. In such settings, an individual wind turbine may make up a significant portion of the total generation, and other generators may need to be blackstarted to pick up load when a wind turbine faults or reduces power. As such, providing the grid operator more time to react to a potential wind turbine fault is critical to prevent load loss. Spinning reserves in microgrids are often diesel
generators, run constantly and inefficiently at a low load fraction. This is a significant microgrid cost and source of pollution. Displacing fossil-fuel generation with renewable generation is often more economically significant in an isolated grid due to higher fuel costs, and fixing a broken wind turbine may be far more temporally and financially expensive due to long supply chains. Finally, the ability to easily implement and modify a control module is a boon to remote systems, where procuring new fault-tolerant wind turbines is expensive.
The goal of this study is to illustrate FIRC functionality under various types of faults and highlight potential wind turbine and grid benefits. Effects of wind turbine control on grid stability are not considered here and will be a topic of future research. Also, the quantitative benefits of an FIRC require its deployment in real-world systems, and will be a topic of future research.

The organization of the article is as follows: Section 2 describes the building blocks of the proposed wind power plant FIRC. Section 3 shows how the proposed controller module can be implemented in both a MATLAB Simulink model and real turbine
control program written in LabVIEW. Section 4 presents the numerical simulations and illustrates the benefits of the controller. Section 5 draws conclusions and lays a road map for future related works.

## 2   Fault Impact Reduction Control Details

The proposed FIRC continuously monitors the wind turbine for various considered faults, mitigates said faults, and reports the state of the wind turbine along with its predicted action to the grid operator. The state of the wind turbine (whether it is
operating normally, experiencing warnings, or shutting down) is described in modes. Wind turbine modes are communicated in the bits shown in Table 1. Multiple modes may be active at once, so multiple bits may be set at once. Normal stop is the most common, gentle type of stop. The blades pitch to slow down, and then the parking brake is applied. Open loop stop occurs when the control system's status is suspect, and the turbine stops using mechanical systems. Emergency stop is the most severe and rapid type of stop: the parking brake is applied, and the blades are pitched to slow down. An emergency stop can cause
high loads, wear, and heat on components, so is undesirable unless absolutely necessary.

### 2.1   Warning Logic

The FIRC checks wind turbine sensors for warnings. Warnings are created for existing fault parameters that would result in a wind turbine stop, with widened thresholds. As such, a warning is triggered before a fault, giving the (micro)grid controller time to react. Warning thresholds were determined somewhat arbitrarily for the demonstration purposes of this study, and



**Table 1.** Wind turbine controller modes. When multiple modes are active, they are added together. For example, if wait and derate warning modes are active, then the binary mode number is [000011], with a mode value of three.

| Mode Value | Mode Definition | Mode Bit |
|:---:|:---:|:---:|
| 0 | No warning | [000000] |
| 1 | Wait warning | [000001] |
| 2 | Derate warning | [000010] |
| 4 | Stop warning | [000100] |
| 8 | Normal stop | [001000] |
| 16 | Open loop normal stop | [010000] |
| 32 | Emergency stop | [100000] |

will depend on the specific needs of the user. Ideally, warning thresholds will be data-driven, and refined based on operation of a real wind turbine equipped with an FIRC module. The uncertainty and noise of the input signals are considered by the wind turbine's FDI system, external to the FIRC. If a fault threshold is reached, the wind turbine stops. Note that some fault parameters are binary and cannot be given warning functionality, meaning, either the parameter is 'faulted' or 'not faulted'. A warning will either trigger stop (SW), derate (DW), or wait (WW) warning mode bits. These three warning modes are described

below. Multiple warnings of a given type may be detected, but only one is required to activate the warning mode. If there are no warnings of a certain type, that warning mode bit is reset.

1. 'Stop': If a stop warning is detected, the stop warning mode bit is set and the stop timer is started, showing how long the warning is active. This warning type indicates that a sensor is approaching a value that will trigger a fault. When no stop warnings are detected, the stop timer and stop warning bit are reset. An example of this warning type is a rotation speed

error, likely caused by a power electronics or sensor failure.

2. 'Derate': If a derate warning is detected, the derate warning mode bit is set and the derate countdown timer is started. The start time, $t_{start}$, will vary by wind turbine, according to the user's specification. If the warning persists for $t_{start}$ time, the timer reaches zero and the turbine maximum power command is decreased to $\frac{1}{2} pu$ (per unit) rated power to reduce stress on the wind turbine, and hopefully mitigate the warning. As long as the warning persists, the power command

remains at $\frac{1}{2} pu$. If this is successful in removing the warning, the controller then tries to maximize turbine power while preventing the warning from reoccurring. The method is thus: If the warning mode is triggered on/off, the timer is reset. If the timer counts down to zero in derate or no warning mode, the power command steps down or up, respectively. The step magnitudes start with the initial $\frac{1}{2} pu$ step down, and halve in magnitude from there: $\frac{1}{4}$, $\frac{1}{8}$, $\frac{1}{16}$, and $\frac{1}{32} pu$. From here on out the step magnitudes stays constant. The power command is limited to $[\frac{1}{2}, \frac{31}{32}] pu$. The controller does not return

to the nominal power command until the operator deals with whatever caused the initial derate warning and resets the FIRC module. This warning type indicates that a sensor is approaching a value that will trigger a fault. An example of

**Figure 2.** Warning logic flow diagram. Mode, timer, and predicted derate (logic not shown) are communicated to the grid. Predicted derate is meaningful once there is a derate warning, and is set to the derate value that will occur when the derate countdown hits zero. First, the controller checks for active warning modes. Then, for each warning mode active, the logic depends on the type of warning mode. If wait or stop, a timer is incremented to track how long it persists. Otherwise, it is reset. If derate, a countdown is started and decremented as long as the derate warning mode persists. If it reaches zero, the power command is halved. If the derate warning mode is exited, the controller seeks the highest power command it can operate at while staying below rated power and greater than or equal to $\frac{1}{2} pu$. The method is: If the derate countdown hits zero in derate warning mode, power decrements. If it hits zero out of derate warning mode, power increments, saturated to $[\frac{1}{2}, \frac{31}{32}] pu$. Steps are progressively halved in magnitude to seek the optimal power command.

this warning type is a high power reading, which could be mitigated by derating the turbine. Figure 5 and Fig. 7 show examples of a wind turbine in derate warning mode.




3. 'Wait': If a wait warning is detected, the wait warning mode bit is set and a the wait timer is started, showing how long the warning has been active. This warning indicates a sensor that will probably not trigger a fault, but requires maintenance. When no wait warnings are detected, the timer and mode bit are reset. An example of this warning type is torque sensor drift.

The flow diagram in Fig. 2 illustrates the warning logic. Results are then communicated to the grid operator. Because multiple modes may be active, the grid operator may see multiple active modes and multiple timers. The presence of one warning mode does not reset another warning mode. However, the presence of a stop mode resets all warning modes, because the turbine is entering shutdown. For instance, if derate and stop warning modes are active due to derate and stop warnings, the binary mode number is [000110]. The derate countdown will decrement from its initial value, and the stop timer will increment from zero. If multiple modes are active, the most power-limiting mode dominates, meaning, when the derate countdown reaches zero, the power command will be limited to $\frac{1}{2} pu$ due to the derate mode. If some time later the warning causing the stop warning reaches the fault threshold, the wind turbine will enter the appropriate stop mode (say, normal stop), the derate and stop warning mode bits will reset, and the wind turbine will shut down. The binary mode number will then be [001000].

## 2.2 Grid Communication

For each wind turbine in a grid, the grid controller needs to know:

1. Which operational mode or modes the wind turbine is in

2. If the wind turbine is in a warning mode, the time until an event occurs

3. The power that will be targeted when said event occurs.

The FIRC sends that information to the system controller: a binary number with a bit representing each mode, the time until power reduction, and the power that will be targeted upon power reduction. The grid operator can reset the FIRC to clear warnings and restore a nominal power target as desired.

## 2.3 Faults Considered

The faults in Table 2 are checked at each time step by the FIRC. They are taken directly from the wind turbine described in Sect. 3.1. The faults checked by this wind turbine are tailored to the needs of the system, and were created by the original control program designers (at this time, this is undocumented). This illustrates one benefit of the FIRC: the ability to easily add new warnings to check with whatever thresholds desired. The selection in this study was chosen to illustrate faults that are easy to demonstrate and trigger a variety of warning types, not based on how often said faults occur. The goal is demonstration, not quantification. A validation campaign with this FIRC would be required to begin to quantify how often faults occurred, and the benefit of the FIRC to the wind turbine. The appendix contains a more detailed table (Table A1) with warning and fault thresholds.





**Table 2.** Wind turbine faults considered, from the NREL three-bladed Controls Advanced Research Turbine (CART3) controller. Warning: the associated warning type we created for the fault. Stop: the type of shutdown that occurs upon fault. HSS: high-speed shaft. LSS: low-speed shaft. MET: meteorological.

| Name | Description | Warning | Stop |
|---|---|---|---|
| GENOVER TEMP | High generator temperature | derate | normal |
| GEARBOXFAIL | High gearbox temperature | derate | normal |
| HIGHXACCEL | High turbine x acceleration | stop | normal |
| FREQSENSORFAIL | Generator and HSS speed disagree (power electronics or cabling or sensor issue) | stop | normal |
| TORQUE SENSOR FAILURE | HSS torque disagrees with torque command or LSS torque | wait | normal |
| GBOILPRESSURELOW | Low gearbox oil pressure | wait | normal |
| YAWBRAKEERROR | Yaw drive hydraulic pump malfunction: high yaw brake oil pressure, yawing when brake set, or yaw brake set unintentionally | wait | normal |
| YAWPOSERROR | Nacelle vane and wind direction disagree. | wait | normal |
| TORQUE OVERLOAD | High HSS/LSS torque | derate | normal |
| HIGHANGULARRATE | High pitch, yaw, or roll acceleration | stop | normal |
| OVERSPEED NORMAL | High HSS/LSS speed | derate | normal |
| MET TEMP PRES FAULT | MET pressure or temperature out of bounds, or MET temperatures different | wait | normal |
| RPMFAIL | HSS and LSS speed disagree | stop | open-loop |
| PEOVERPOWER | Instant, 1-s-average power, or 1-min-average power above bounds | derate | emergency |

## 3 Simulation Setup

In this study, we performed simulations to demonstrate the FIRC functionality described in Sect. 2 using MATLAB Simulink. The model comprises a wind turbine equipped with an FIRC module. Power flow calculations demonstrate the benefit of having an FIRC-equipped wind turbine connected to a utility grid and a microgrid under a variety of fault scenarios, compared to a baseline turbine which behaves at the industry standard: giving no fault warnings and immediately entering shutdown when a fault threshold is crossed.

### 3.1 Wind Turbine Model Setup

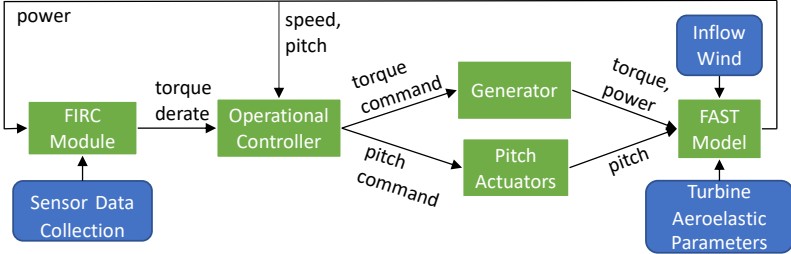

**Figure 3.** Simplified CART3 model block diagram.



The wind turbine modeled is the three-bladed Controls Advanced Research Turbine (CART3), a recently-decommissioned 600 kW machine previously located at NREL's Flatirons Campus in Boulder, CO. Information on the CART3 and its commissioning was documented by Fingersh and Johnson (Fingersh and Johnson, 2002). We developed the original FIRC logic in the CART3 controller, which is written in LabVIEW. Figure A1 in the appendix details a section of the LabVIEW imple-
mentation, which included the existing fault safety control, SCADA system, and operational controller, and has a variety of differences with the Simulink implementation. For instance, in the Simulink implementation, torque was controlled to change output power, whereas in the LabVIEW implementation, rotation speed was controlled. Despite the differences, the FIRC module was simple to transfer between implementations, which bodes well to future implementation in commercial machines. Hardware-in-the-loop simulations with the physical SCADA system and a simulated wind turbine were run in the CART3
system, and performed as desired.

For the Simulink implementation, the CART3 is modeled using the FAST code developed at NREL, and compiled for use in Simulink using an S-function. At this time, no documentation of the baseline Simulink model, developed previously at NREL and modified for this study, is available. A realistic, turbulent inflow wind file is used for each simulation. The full model can be found on GitHub (Anderson, 2021). With this model, all scenario results can be replicated, new warning/fault checks can be
easily added, and new scenarios can be run. A simplified model schematic is displayed in Fig. 3. The FIRC module derates the wind turbine's output power via torque control. The original, operational controller controls power by regulating torque below rated speed, and pitch above rated speed: The torque command follows a traditional optimal torque-speed curve in region 2, a linear torque-speed curve in transition region 2.5, and is set to a maximum torque in region 3. Pitch is controlled by a PI controller tracking rated speed. The generator and pitch actuators try to track the given torque and pitch commands, and feed
generator torque, power, and blade pitch angle into the FAST model, which outputs parameters such as rotor speed and loads.

### 3.2 Microgrid Calculation Setup

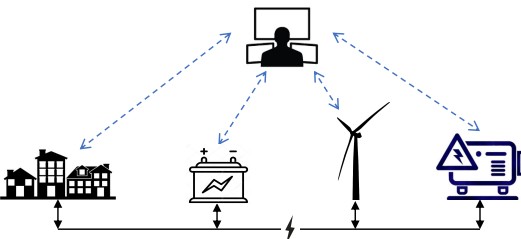

**Figure 4.** Grid calculation. System controller receives information from all nodes and gives commands to generation.

The microgrid calculation used for Scenario 2 and Scenario 3 in Sect. 4 comprises a varying load powered by a wind turbine, battery bank, diesel genset, and microgrid controller. Figure 4 highlights these main sections of the grid calculation. The mean load is 95 kW, the turbine rated power is 100 kW, the diesel rated power is 100 kW, the battery charge/discharge capacity is 100 kW, and the battery storage capacity is 10 kWh. The calculation considers a microgrid controller that reads the load,





generation, and information from the FIRC, and dispatches generation to maintain load while maximizing wind contribution and battery charge. The battery smooths the difference between load and wind/diesel generation. Note that this is a calculation to demonstrate various scenarios, not a model.

### 3.3 Demonstration Cases

We perform the following Simulink simulations described in Table 3 to demonstrate the operation of the FIRC, the interaction between different warnings, and the information communicated to the grid.

**Table 3.** Simulation cases demonstrating FIRC module operation.

| Case | Warning Modes Included | Purpose |
|------|------------------------|---------|
| 1 | Derate | Demonstrate FIRC operation with a single derate warning, where derating mitigates the warning |
| 2 | Wait, Stop | Demonstrate FIRC operation with multiple types of warnings, where a fault occurs after a corresponding stop warning |
| 3 | Derate (two) | Demonstrate FIRC operation with multiple derate warnings |

### 3.4 Demonstration Scenarios

The following scenarios described in Table 4 are evaluated to demonstrate the potential grid benefits of a wind turbine equipped with an FIRC, compared to a baseline controller. These are calculations, not simulations.

**Table 4.** Scenarios demonstrating FIRC benefit over a baseline controller.

| Scenario | Description | Purpose |
|----------|-------------|---------|
| 1 | Grid-connected; wind turbine experiences high generator temperature at night | Demonstrate power reduction mitigating error and preventing shutdown |
| 2 | Microgrid; wind turbine experiences high accelerometer reading resulting in fault | Demonstrate stop warning softening turbine shutdown: preventing diesel emergency start, reducing maximum power draw from storage, and decreasing risk of transient grid instability |
| 3 | Microgrid; wind turbine experiences high accelerometer reading not resulting in fault | Demonstrate case where diesel generator is brought online and not needed, but sensor drift is revealed |





## 4 Results


### 4.1 Demonstration Cases

#### 4.1.1 Case 1: Derate Warning

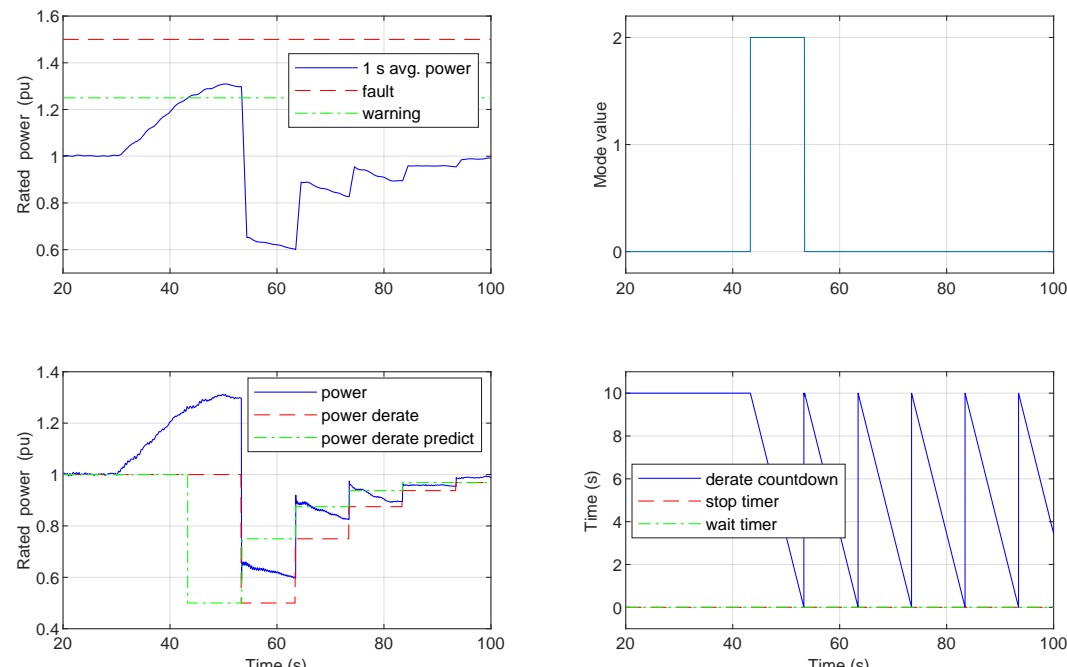

**Figure 5.** Case 1 scope. Overpower warning mitigated by derate. Power then progressively steps up to near 100% ($\frac{31}{32}\,pu$), with power staying below the warning threshold. Top left: 1 s avg. power signal with fault and warning thresholds. Bottom left: power prediction, command, and actual output. Top right: mode value, two=derate warning, zero=no warning. Bottom right: timers. In derate warning mode, derate countdown counts down from the user-defined max time. Each time it reaches zero, power command steps down if in derate warning mode or up if not in derate warning mode.

Case 1, shown in Fig. 5, demonstrates FIRC operation with a single derate warning, where derating mitigates the warning. When 1 s average power exceeds its warning threshold at 43 s, derate warning mode is entered (mode value increases by two), and the power prediction is set to $\frac{1}{2}$ per unit (pu) of rated power because the power will be derated to that value if the warning persists. The derate countdown is started. When it reaches zero at 53 s, the derate occurs as predicted. The derate reduces 1 s average power and mitigates the warning, returning the turbine to no warning (mode value decreases by two). At this time, the power predicted is set to $\frac{3}{4}$ pu, as the wind turbine will increase its power rating if a warning does not resurface. The derate countdown resets and begins a new countdown. When it hits zero, the power increase occurs, and the process repeats. Step changes in power command are halved with each step, reaching a maximum of $\frac{31}{32}$ pu. Power is not returned to 1 pu, as operation at this point led to a warning initially. If a warning were to resurface, the power command would step down. The







grid operator can reset the warning mode at any time to return the wind turbine to normal operation. This would be useful in a situation where the cause of the overpower warning is temporary, such as a series of abnormally high gusts.

### 4.1.2 Case 2: Derate Warning and Stop Warning Predicting Normal Stop

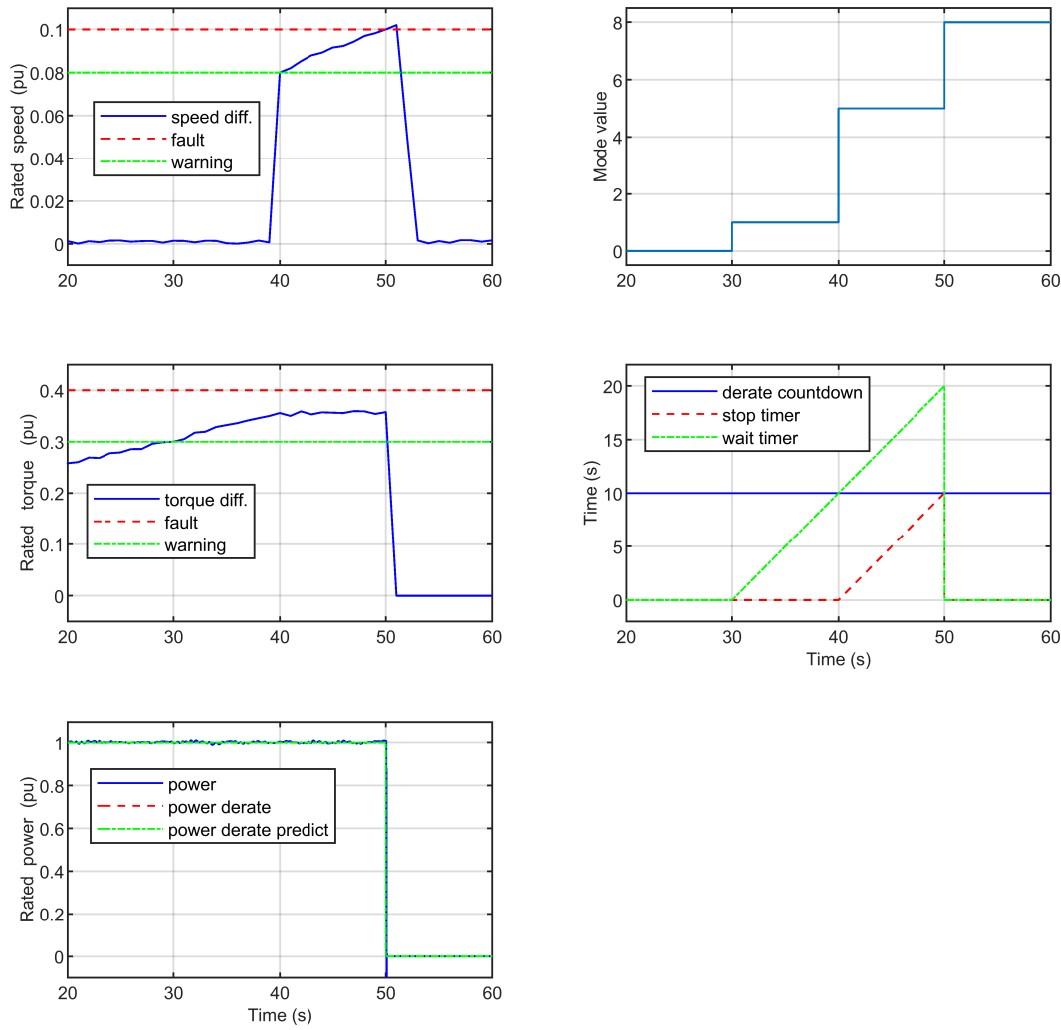

**Figure 6.** Case 2 scope. Speed sensor warning anticipates fault, which would give the grid operator time to react. Top left: HSS/LSS speed difference, with fault and warning thresholds. Mid left: HSS/LSS torque difference, with fault and warning thresholds. Bottom left: power prediction, command, and output. Top right: mode value. 0=no warning, 1=wait warning, 5=wait warning + stop warning, 8=normal stop. Mid right: timers. Stop and wait timers start counting up when stop and wait warning modes are activated, respectively, then reset when the wind turbine shuts down.





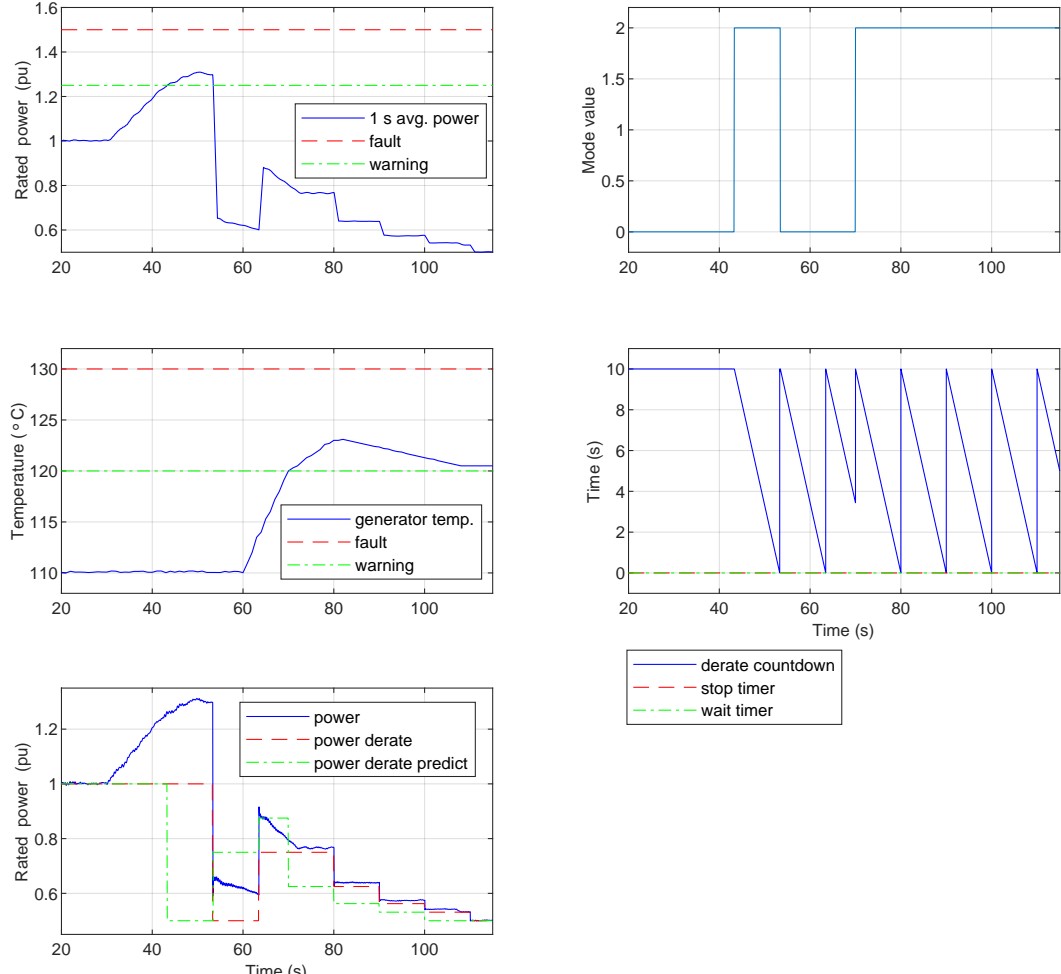

**Figure 7.** Case 3 scope. Interaction between two derate warnings. Derating mitigates the first warning, and the power command begins to step up, but the appearance of a second warning drives the power command to step back down. Top left: 1 second average power, with fault and warning thresholds. Mid left: generator temperature, with fault and warning thresholds. Bottom left: power prediction, command, and output. Top right: mode value. 0=no warning, 2=wait warning. Mid right: timers. In derate warning mode, derate countdown counts down from the user-defined max time. Each time it reaches zero, power command steps down if in derate warning mode or up if not. The timer resets when derate warning mode toggles on/off.

Case 2, shown in Fig. 6, demonstrates FIRC operation with two different warning types, wait and stop, where a fault occurs after a corresponding stop warning. The difference between the HSS and LSS torque sensor exceeds the warning threshold at 30 s, and wait warning mode is entered (mode value increases by one). The wait timer begins to count up, showing how long the warning has persisted. At this point, the grid operator could decide to send a technician to examine the warning's source. At 40 s, the difference between the HSS and LSS speed sensors exceeds the warning threshold, and stop warning mode is entered





(mode value increases by four). The stop timer begins to count up, showing how long the warning has persisted. At this point, the grid operator could bring other generation online in case of an impending wind turbine shutdown. At 50 s, the difference between the HSS and LSS speed sensors exceeds the fault threshold, and a fault is triggered, causing the turbine to normal stop and exit all other modes (mode value = 8). The power command is set to zero, and all timers are reset. Because of the FIRC, this abrupt stop is anticipated, and the grid operator would likely have time to bring compensating generation online.

### 4.1.3 Case 3: Derate warning from multiple parameters

Case 3, shown in Fig. 7, demonstrates FIRC operation with multiple derate warnings. It begins the same as Case 1: when 1 s average power exceeds its warning threshold at 43 s, derate warning mode is entered (mode value increases by two) and the power prediction is set to $\frac{1}{2}$ pu. The derate countdown is started. When it reaches zero at 53 s, the derate occurs as predicted. The derate mitigates the warning, returning the turbine to no warning mode (mode value decreases by two), and
power prediction steps up to $\frac{3}{4}$ pu. The power steps up to the predicted value when the derate countdown hits zero at 63 s, but before it can step up again, derate warning mode is again triggered by a different warning (mode value increases by two) when the generator temperature exceeds its warning threshold at 70 s. This resets the derate countdown, and its persistence leads to power command step-downs of decreasing size, with the power command finally reaching the lower limit of $\frac{1}{2}$ pu. The lower limit is set to prevent model controller instability.

### 4.2 Demonstration Scenarios

The following scenarios demonstrate the benefit of an FIRC compared to a baseline wind turbine controller (from a grid perspective).





### 4.2.1 Scenario 1: Derate Prevents Fault in Grid-Connected Wind Turbine

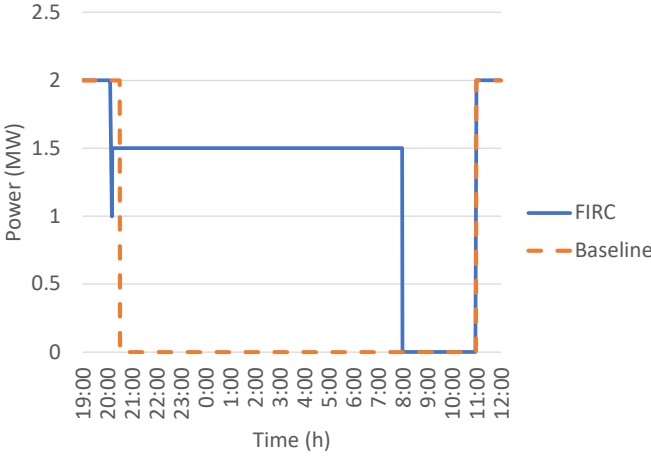

**Figure 8.** FIRC derates wind turbine to avoid fault, increasing wind power plant production by 17 MWh from time of high sensor readings to time of repair.

In this scenario, a 2 MW, grid-connected wind turbine is operating at rated power under high winds. Figure 8 shows the
power output of a baseline and FIRC-equipped wind turbine throughout the scenario. At 20:00, high winds combined with a malfunction in the cooling system lead to increasing generator temperature. In the baseline wind turbine case, at 20:30, the wind turbine faults as the temperature sensor crosses the fault threshold, and it shuts down immediately. In the FIRC wind turbine case, at 20:00, the wind turbine enters derate warning mode. It tells the grid that it will derate to 50% at 20:10 unless the warning resolves. The readings continue to increase, and the wind turbine reduces power to 50% at 20:10, as predicted.
The derated operation reduces generator temperature and mitigates the warning, and the turbine increases power to 75% at 20:20. The controller finds this to be the maximum operating point while staying below the warning threshold, so the turbine remains at this operating point. In both cases, a technician is sent to repair the cooling system at 08:00 the following morning and completes the repair in 3 hours. At 11:00, the wind turbine resumes operation at rated power. In this scenario, the large utility grid easily absorbs the loss or derate of a single wind turbine. However, the FIRC maximizes energy production. From
the time of the high sensor readings (20:00) to when the turbine is repaired and resumes operation (11:00 the next day), the baseline wind turbine produces 1 MWh, whereas the FIRC-equipped wind turbine produces 18 MWh. Finally, the damage potential to the wind turbine because of high generator temperature is minimized, by derating instead of running until fault. This scenario highlights the value of FIRC in conventional applications. To understand the life-cycle financial impact of the FIRC module, in terms of maintenance costs, power production, LCOE, etc. will require long-term validation exercises of wind
turbines equipped with FIRCs and operating in various conditions, both grid-connected and isolated. This is a topic of future research.





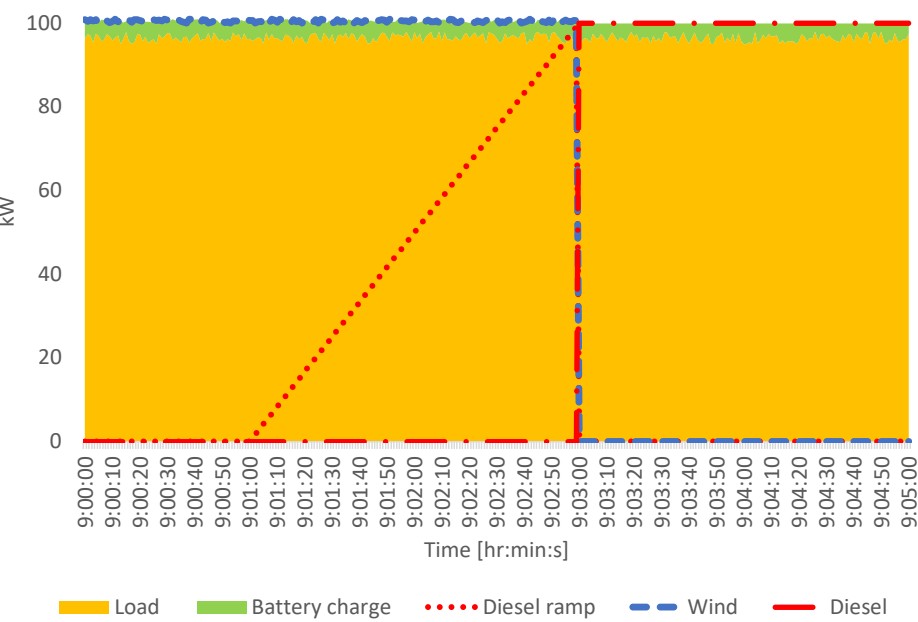

**Figure 9.** Scenario 2 baseline case. Abrupt wind turbine fault necessitates diesel emergency start and maximum battery discharge.

### 4.2.2 Scenario 2: Stop Warning Eases Microgrid Generation Transition

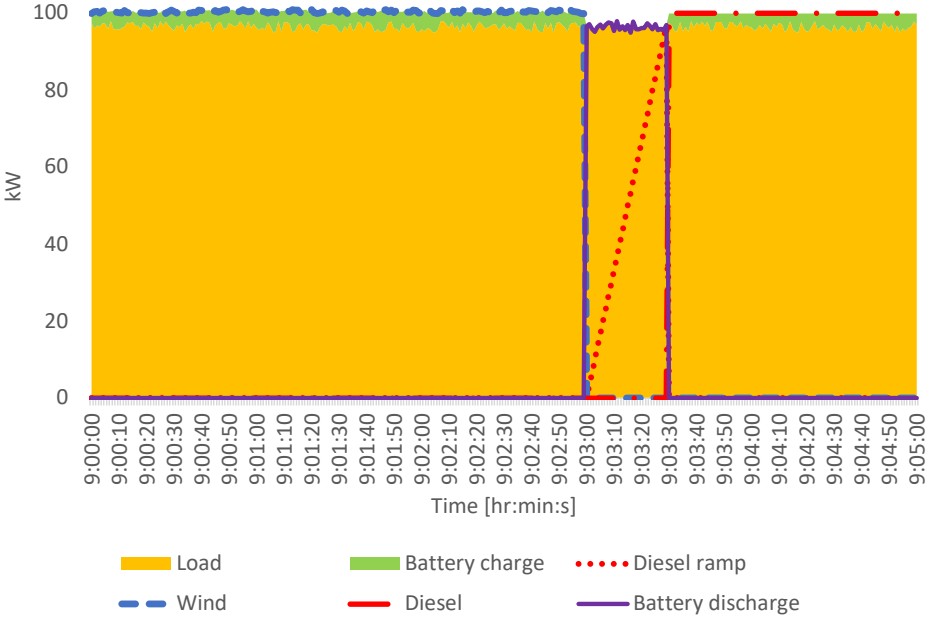

**Figure 10.** Scenario 2 FIRC case. Anticipated wind turbine fault allows for a seamless transition from wind to diesel generation, without battery discharge.




In this scenario, a microgrid comprises a 100 kW peak load, a 100 kW wind turbine, a 100 kW diesel genset, and a 100 kW, 10 kWh lithium-ion battery bank (as mentioned in Sect 3.2). The battery charges or discharges to smooth the difference between the variable load and generation. At 09:00, under high winds, an accelerometer begins to show high readings.

With a baseline wind turbine (see Fig. 9), at 09:03 the wind turbine faults. The battery is forced to provide near-maximum power to support the grid, and the diesel genset emergency starts. Once the diesel ramps up, it replaces the battery as the generation source, and recharges it.

With an FIRC (see Fig. 10), at 09:01 the accelerometer crosses a warning threshold, and the wind turbine enters stop warning mode, warning the microgrid controller that a fault is imminent. The microgrid controller responds by ramping up the diesel at a relaxed rate. At 09:03, the wind turbine faults, and the ready, ramped-up diesel is connected to provide the needed generation, with the battery continuing to provide smoothing. Compared to the baseline controller, the FIRC benefits the microgrid in a variety of ways: it prevents the battery from experiencing both maximum power charge/discharge (8% vs. 100%) and a low state of charge, thus increasing battery longevity. It prevents emergency-starting the diesel, which is inadvisable unless the diesel is already warm and running, and reduces the risk of the diesel not synchronizing when it connects to the grid, which could cause the whole grid to crash. Diesel ramp rate is reduced from 200 kW min$^{-1}$ to 50 kW min$^{-1}$. Overall, it decreases the risk of transient grid instability and load loss by allowing a planned shift in generation, thus increasing the microgrid's resilience under transitions.

### 4.2.3 Scenario 3: Fuel Use vs. Information Trade-Off

This scenario is a variant of Scenario 2. The sole difference is that the accelerometer reading levels out below the fault threshold. With a baseline controller, normal operation continues, as the fault threshold is never crossed. With an FIRC, the diesel genset may be started but not needed. The grid operator has the choice to shut down the wind turbine and troubleshoot the high accelerometer reading, or to reset the FIRC warning and continue normal operation. In this scenario, the FIRC may cause more diesel use, (the amount will depend on the operator's course of action) but provides the grid operator with a warning of a potential issue developing in the wind turbine. In a microgrid wherein individual generation units are critical to the system, finding and fixing such issues as early as possible is well worth the modest extra operational expense. Note that if this event occurred in a grid-connected scenario, the FIRC would provide the benefit of information without costing the grid anything, as the utility grid would not have to respond to the prospect of an individual wind turbine faulting.

### 5 Conclusions

This article presents a novel fault impact reduction control (FIRC) module which could be integrated or implemented into existing wind turbine controls to mitigate faults and communicate predicted wind turbine actions to the grid operator. This functionality is especially beneficial to microgrids and isolated grids, in which a single wind turbine may provide a large portion of the generation capacity. We perform simulations to demonstrate the warning logic of the FIRC, along with its communication to the grid operator. Various scenarios highlight the benefit of an FIRC-equipped turbine over a baseline machine in both





utility grids (Scenario 1) and microgrids (Scenario 2): maximizing wind turbine power production, minimizing downtime, and facilitating transitions between generation sources in the face of a fault. In Scenario 1, from the time of sensor malfunction to repair the FIRC-equipped wind turbine produces more energy (18 MWh vs. 1 MWh) and has less downtime (3 h vs 14.5 h). In Scenario 2, the FIRC-equipped wind turbine reduces maximum battery charge/discharge from 100% to 8%, and reduces diesel ramp rate from 200 kW min$^{-1}$ to 50 kW min$^{-1}$.

Future work includes both controller development and demonstration. Different warnings and warning thresholds can be added to the flexible framework of the presented controller module, which itself can be easily integrated into existing controllers on deployed machines. Different warning modes can be implemented to suit various grid and wind turbine needs, and to accommodate faults that could be ameliorated by actions besides power reduction. For example, the pitch fault tolerant control that Liu et al. (2021) proposed could be integrated into the FIRC framework. Faults checked and warning logic should be
tailored to the SCADA system of a specific machine. Although high level fault data exists for wind turbines as a whole, this needs refining to different wind turbine classes, sizes, and models. Partnership with wind turbine manufacturers could help determine which faults are most common for a specific machine, the optimal control action to ameliorate them, and the best way to integrate a FIRC module into their designs. Partnership with communities that employ distributed wind turbines, both in isolated grid, microgrid, and utility grid contexts, could help determine which faults are most common to their specific use
case, provide experience integrating an FIRC module into an existing, operational system, and measuring its impact in that system. Such real-world integration could validate the function and quantify the resilience and financial benefits of the FIRC module. More advanced two-way communication from grid operator to wind turbine could be implemented to optimize control actions. Faults in the grid could be considered; for example, the turbine could momentarily disconnect from a faulting grid, and attempt to reconnect a short time later. If the fault still persists, the turbine could repeat this a few times, until ultimately
reconnecting to a stable grid or shutting down.





## Appendix A

**Table A1.** Detailed information on quantities checked and their thresholds.

| Name | Description | Warning threshold | Fault threshold | Warning type | Stop type |
|---|---|---|---|---|---|
| GENOVER TEMP | Generator temperature > x°C | 120 | 130 | derate | normal |
| GEARBOXFAIL | Gearbox temperature > x°C | 70 | 80 | derate | normal |
| HIGHXACCEL | Turbine x accelerometer > x g | 0.4 | 0.5 | stop | normal |
| FREQSENSORFAIL | Generator and HSS speed disagree by > x (power electronics or cabling or sensor issue) | 8% | 10% | stop | normal |
| TORQUE SENSOR FAILURE | HSS torque disagrees with torque command by > x or LSS torque by > y | 15% 30% | 20% 40% | wait | normal |
| GBOILPRESSURELOW | Gearbox oil pressure under x Pa | 1100 | 1000 | wait | normal |
| YAWBRAKEERROR | Yaw drive hydraulic pump malfunction. (Yaw brake oil pressure > x Pa, yawing when brake set, or yaw brake set unintentionally.) | 1900 | 2000 | wait | normal |
| YAWPOSERROR | Nacelle vane + yaw position ≠ wind direction OR nacelle vane - wind direction > x° | 35 | 45 | wait | normal |
| TORQUE OVERLOAD | HSS/LSS torque > x rated | 140% | 150% | derate | normal |
| HIGHANGULARRATE | Pitch acceleration > x m s$^{-2}$, yaw acceleration > y m s$^{-2}$, or roll acceleration > z m s$^{-2}$ | 4, 3, 3.5 | 5, 4, 4.5 | derate | normal |
| OVERSPEED NORMAL | HSS/LSS speed > x*rated | 105% | 110% | derate | normal |
| MET TEMP PRES FAULT | MET pressure outside of [a, b] kPa, or temperature outside of [-c, c]°C, or MET temperatures > d°C different | 105, 55, 35, 10, | 110, 50, 40, 15 | wait | normal |
| RPMFAIL | HSS and LSS speed > x different | 8% | 10% | stop | open loop |
| PEOVERPOWER | Instant power > x rated, 1 s average power > y rated, or 1 min average power > z rated | 142% 125% 103% | 183% 150% 105% | derate | emergency |



**Figure A1.** A sample from the FIRC implementation in the CART3 controller, written in LabVIEW. The controller loops through various fault checks. In this sample, it is performing the TORQUE SENSOR FAIL check. If warning conditions exist, it sets the wait mode bit and increments the wait timer. Otherwise, it zeroes the wait mode bit and the wait timer.

*Code and data availability.* Code and data are publicly available at the following github repository: https://github.com/badeshiben/Fault-Impact-Reduction-Control-Module

*Author contributions.* Benjamin Anderson developed the fault impact reduction control module in labVIEW and then in Simulink, integrated
it into the Simulink wind turbine model, ran the Simulink simulations and other calculations, and wrote the paper. Edward Baring-Gould created the idea of a fault impact reduction control module, helped Benjamin develop its logic, and edited the paper.



*Competing interests.* The authors declare that they have no conflict of interest.

*Acknowledgements.* The authors would like to thank Jim Reilly, Ram Poudel, Venkat Krishnan, Robert Preus, Daniel Zalkind, and Lee Fingersh for their thoughtful reviews. The fault impact reduction control module was developed under the U.S. Department of Energy Wind

Energy Technologies Office's Microgrids, Infrastructure Resilience, and Advanced Controls Launchpad (MIRACL) project. This work was authored by the National Renewable Energy Laboratory, operated by Alliance for Sustainable Energy, LLC, for the U.S. Department of Energy (DOE) under Contract No. DE-AC36-08GO28308. Funding provided by the U.S. Department of Energy Office of Energy Efficiency and Renewable Energy Wind Energy Technologies Office for the Microgrids, Infrastructure Resilience, and Advanced Controls Launchpad (MIRACL) project. The views expressed in the article do not necessarily represent the views of the DOE or the U.S. Government. The U.S.

Government and the publisher, by accepting the article for publication, acknowledges that the U.S. Government retains a nonexclusive, paid-up, irrevocable, worldwide license to publish or reproduce the published form of this work, or allow others to do so, for U.S. Government purposes.



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
