# Peer review of "Demonstration of a Fault Impact Reduction Control Module for Wind Turbines"

_Wind Energy Science, 2022_

## Referee Comment (RC1)

**Review for WES-2022-27**

**General assessment**

**Technology**

This paper presents a novel Fault Impact Reduction Control (FIRC) module. Key advances are the potential prevention of an impending fault by a derating strategy, and the communication of useful turbine information (warning mode, warning time, predicted time until derating step, predicted derated power) to the grid operator. Furthermore, the FIRC can be deployed as an add-on to existing operational controllers and fault detection logics. The FIRC can be very helpful in rendering wind turbines more cooperative and operator-friendly.

**Contribution**

The main contributions of this paper are from my point of view:

- Very comprehensive overview of fault detection, isolation and control. Thank you very much for that!
- Development of a solid base concept for FIRC, which could stimulate various future research (for instance, communication of probabilities of actual stop within the next 1s/10s/100s to the grid operator).
- The task and solution seem to be very relevant for and very close to industrial application.
- Very thorough and clear explanation of the FIRC logic.
- Solid and successful demonstration of the FIRC behavior under various fault cases, and of the FIRC effects in national grid or microgrid scenarios.

**Reservation**

In the abstract you state "controls to ameliorate such faults are *uncommon* in research and industry". Unfortunately, I cannot judge to what extend pre-fault warnings and preventive derating could already be state-of-the-art in the wind industry.

**Proposed minor corrections**

The following set of proposed minor corrections may improve the already very solid paper:

- Abstract, line 1: Does "in distribution applications" mean that the turbines are connected to the distribution grid, or that they are distributed around a wider area, i.e. are not part of a wind farm?
- Abstract, line 12: A limitation to "easy to integrate with the existing controller" may be, that an existing commercial operational controller would be required to have a derate functionality and an input port for derating commands.
- Section 1, line 22: The term probably should be "shutdowns" instead of "shutdown".
- Section 1.2, line 58: The "Pourmohammed" reference contains one surplus bracket.
- Section 1.2, line 85: It could be specified that the stated goals are only two amongst many others (e.g. also alleviation of tower loads).
- Section 2, line 138: Maybe you can specify which "mechanical systems" are utilized in an open loop stop.
- Section 2.1, Derate warning mode: I understood that the unit "pu" in the power context is always with reference to rated power. If so, I am not sure about the fixed upper saturation limit of 31/32pu. Shouldn't the upper limit be dynamically set to one power step below the

lowest power where a derate warning has been triggered since the last FIRC reset by the grid operator? Example: A derate warning is triggered at P_derate = 1pu, but is also present when P_derate =31/32pu. The FIRC steps down to P_derate = 0.5pu and the derate warning vanishes. Consequently, the FIRC would gradually step up until 31/32pu, where the derate warning is triggered again. Consequently, the procedure of stepping down and back up to 31/32pu would be repeated, right? Thus, wouldn't P_derate periodically oscillate between 31/32pu and 30/32pu, and periodically activate the derate alarm? The above proposed dynamic rule would set an upper saturation of 30/32pu = 31/32pu (lowest power with derate warning) – 1/32pu (smallest step size), preventing this oscillation.

- Section 2.1, Stop warning mode: Could it be reasonable to also inform the grid operator about the expected normal/open-loop/emergency stop power trajectory? Depending on the ramp slope, the grid operator could select the best spinning reserve generator.
- Section 2.1, lines 168 f.: The jump to Fig. 5 and 7 in the results section feels a bit distractive and does not seem to be necessary.
- Section 2.3, Table 2: Which acceleration quantity (location, direction) is represented by "x acceleration"?
- Section 4.1, Fig. 5: Why does the actual power seem to have an always-positive steady-state offset to the commanded derate power? Consequently, also the power predictions would be misleading for the grid operator.
- Section 4.1, Fig. 6: Why is the actual power negative at t=50s?
- Section 4.1.3, line 274: Does "model controller instability" refer to instability of the FIRC or of the operational controller?
- Section 4.2.2: I have the impression that the figure files (but not the captions) have been interchanged for Fig. 9 and Fig. 10.
- Section 4.2.2: In the scenario with FIRC, the turbine exactly faults at that moment when the diesel generator has completed its start-up. However, it would be fair to state that this fault event time represents a lucky optimal coincidence, and could have happened either during diesel start-up or hours after diesel start-up.
- Section 4.2.2.: Thinking further, in some scenarios of short time frames between warning and fault and with an expensive battery, it may be advisable to emergency-start the diesel after the FIRC stop warning, right?
- Section 5: For me it would be interesting to extend the outlook by potential benefits of FIRC-equipped turbines in a wind farm control setting. For instance, the derate prediction also represents a prediction of changes in the wakes, which could be utilized to proactively optimize operation of the other turbines in the farm.

---

## Author Response (AR1)

**Reply on RC1**

Benjamin Anderson and Edward Baring-Gould

Thank you for your throughtful review. Our responses are below:

- Abstract, line 1: Does "in distribution applications" mean that the turbines are connected to the distribution grid, or that they are distributed around a wider area, i.e. are not part of a wind farm?
*Thanks, this should have been "distributed applications". Typo corrected. It means the latter.*

- Abstract, line 12: A limitation to "easy to integrate with the existing controller" may be, that an existing commercial operational controller would be required to have a derate functionality and an input port for derating commands.
*True. Updated to "most existing controllers, just requiring derate capability". The updated controller in this study was in view, but I do think it's better to focus on potential applications.*

- Section 1, line 22: The term probably should be "shutdowns" instead of "shutdown".
*Agreed and corrected.*

- Section 1.2, line 58: The "Pourmohammed" reference contains one surplus bracket.
*Corrected*

- Section 1.2, line 85: It could be specified that the stated goals are only two amongst many others (e.g. also alleviation of tower loads).
*Good point; added this.*

- Section 2, line 138: Maybe you can specify which "mechanical systems" are utilized in an open loop stop.
*Updated the stop discussions with relevant details about systems used and ramp rates.*

- Section 2.1, Derate warning mode: I understood that the unit "pu" in the power context is always with reference to rated power. If so, I am not sure about the fixed upper saturation limit of 31/32pu. Shouldn't the upper limit be dynamically set to one power step below the lowest power where a derate warning has been triggered since the last FIRC reset by the grid operator? Example: A derate warning is triggered at P_derate = 1pu, but is also present when P_derate =31/32pu. The FIRC steps down to P_derate = 0.5pu and the derate warning vanishes. Consequently, the FIRC would gradually step up until 31/32pu, where the derate warning is triggered again. Consequently, the procedure of stepping down and back up to 31/32pu would be repeated, right? Thus, wouldn't P_derate periodically oscillate between 31/32pu and 30/32pu, and periodically activate the derate alarm? The above proposed dynamic rule would set an upper saturation of 30/32pu = 31/32pu (lowest power with derate warning) – 1/32pu (smallest step size), preventing this oscillation.
*I've added a footnote discussing this. I've kept the logic as is, as it's simple and maximizes potential power output. It's mostly to demonstrate the concept, and would likely be refined using ML/AI to develop an advanced algorithm.*

- Section 2.1, Stop warning mode: Could it be reasonable to also inform the grid operator about the expected normal/open-loop/emergency stop power trajectory? Depending on the ramp slope, the grid operator could select the best spinning reserve generator.
*That would be helpful information. Added as an output.*

- Section 2.1, lines 168 f.: The jump to Fig. 5 and 7 in the results section feels a bit distractive and does not seem to be necessary.

*Thanks for the feedback. I've removed the references.*

- Section 2.3, Table 2: Which acceleration quantity (location, direction) is represented by "x acceleration"?

*Fore-aft. Added.*

- Section 4.1, Fig. 5: Why does the actual power seem to have an always-positive steady-state offset to the commanded derate power? Consequently, also the power predictions would be misleading for the grid operator.

*This is because of the specific warning in question: the warning is that the 1-s average power is too high, which could happen because of, for example, high turbulent winds causing the wind turbine to produce more power than expected. So, the derate keeps the wind turbine safe, but it still temporarily produces more power than expected.*

- Section 4.1, Fig. 6: Why is the actual power negative at t=50s?

*That's a model artifact, which I have filtered out.*

- Section 4.1.3, line 274: Does "model controller instability" refer to instability of the FIRC or of the operational controller?

*Clarified: "Instability of this model's operational controller".*

- Section 4.2.2: I have the impression that the figure files (but not the captions) have been interchanged for Fig. 9 and Fig. 10.

*Thank you, you are correct. Switched back to the right files.*

- Section 4.2.2: In the scenario with FIRC, the turbine exactly faults at that moment when the diesel generator has completed its start-up. However, it would be fair to state that this fault event time represents a lucky optimal coincidence, and could have happened either during diesel start-up or hours after diesel start-up.

*This is a good point. I've updated the diesel ramping to be more realistic. The diesel synchronizes for 30 seconds, then connects and ramps up. In the FIRC case, as it ramps up, the wind turbine ramps down. I also included the wind turbine ramp down that corresponds to a normal stop in the baseline case.*

- Section 4.2.2.: Thinking further, in some scenarios of short time frames between warning and fault and with an expensive battery, it may be advisable to emergency-start the diesel after the FIRC stop warning, right?

*That's right. There's a tradeoff between the potential of high battery power flow (reduces battery life) and the danger of crash-starting a cold diesel. I've chosen a more relaxed start in the FIRC case.*

- Section 5: For me it would be interesting to extend the outlook by potential benefits of FIRC-equipped turbines in a wind farm control setting. For instance, the derate prediction also represents a prediction of changes in the wakes, which could be utilized to proactively optimize operation of the other turbines in the farm.

*I like this idea. Added as future work.*

[Figure]

**Reply on RC2**

Benjamin Anderson and Edward Baring-Gould

Thank you for your comments. The idea of the FIRC module is to create another safety layer that surrounds the existing fault safety checks, but does not supersede them. This allows the wind turbine to take proactive action to prevent faults, and notify the turbine operator that a fault and shutdown may soon occur. The idea is to demonstrate a control module that would be applicable to a wide variety of wind turbines. In determining the sensors to check and their respective warning modes for this specific wind turbine (CART3), I worked with the senior wind turbine operator/researchers and the existing control programs that they had created. This process will yield different results for different wind turbines. For each machine, the sensors to check and appropriate warning modes to enter may vary based on its physical characteristic and power system application. I have added some language to section 2.3 to this effect.